## RESEARCH ARTICLE

# Loss of sphingomyelin synthase-1 does not cause egg retention or locomotion defects in *Caenorhabditis elegans*

Wenyue Wang*, Xiaoyan Gao and Roger Pocock*

## ABSTRACT

Sphingomyelin is a critical sphingolipid found in plasma membranes of metazoa that provides structural and communicative functions. Sphingomyelin synthases are key enzymes that generate sphingomyelin but their precise functions in animal development and function are not fully understood. The *Caenorhabditis elegans* model encodes five sphingomyelin synthases (*sms-1-5*). Previously, egg-laying and locomotion phenotypes were observed in an *sms-1(ok2399)* deletion mutant. In this study, we attempted to replicate these findings to enable mechanistic dissection of sphingomyelin function. We indeed found that the *sms-1(ok2399)* mutant exhibited egg-laying and locomotion defects, however, we were unable to rescue this phenotype. Further, we generated two additional *sms-1* deletion mutants (*rp398* and *rp399*) and found that their egg-laying and locomotion behaviour is not different to wild-type animals. We suggest that the *sms-1(ok2399)* contains a background mutation that causes behavioural deficits, and that SMS-1 loss does not overtly affect *C. elegans* egg-laying or locomotion.

KEY WORDS: *Caenorhabditis elegans*, Sphingomyelin synthase, Egg-laying, Locomotion

## INTRODUCTION

Sphingolipids are amphipathic molecules with reported functions in controlling cell adhesion and migration, cell death and cell proliferation (Maceyka and Spiegel, 2014). Given these fundamental functions, studies of sphingolipids and their metabolic pathways have extended to physiological and pathological conditions, such as motor dysfunction, insulin resistance, cancer, cardiovascular disorders, and neurological diseases (Hannun and Obeid, 2018; McCluskey et al., 2022; Pan et al., 2023). Sphingomyelin (SM) is a major structural sphingolipid in cellular membranes, composed of a phosphocholine head and a ceramide backbone (Slotte, 2013). SM interacts with cholesterol to form membrane microdomains (lipid rafts), which act as signalling platforms for cellular communication (Chakraborty and Jiang, 2013). In addition, SM metabolism generates bioactive molecules, such as ceramide, that mediate diverse cellular signalling and regulatory processes (Stith et al., 2019).

Development and Stem Cells Program, Monash Biomedicine Discovery Institute and Department of Anatomy and Developmental Biology, Monash University, Melbourne, VIC 3800, Australia.

*Authors for correspondence (roger.pocock@monash.edu; wenyue.wang1@monash.edu)

R.P., 0000-0002-5515-3608

SM levels are tightly controlled by a network of enzymes, including SM synthases and sphingomyelinases, with ceramide acting both as a SM precursor and breakdown product (Kuo and Hla, 2024). Disruption of SM metabolism, often accompanied by altered ceramide levels, has been implicated in motor neuron diseases, where it may contribute to neuronal degeneration by affecting membrane organization, lipid raft-associated signalling, and cellular stress responses (McCluskey et al., 2022). Despite these associations, it remains unclear how individual SM metabolic enzymes specifically influence motor neuron function *in vivo*.

We previous showed that the nematode *Caenorhabditis elegans* is an excellent genetically tractable model to investigate sphingolipid functions (Wang et al., 2023). The *C. elegans* genome encodes five SM synthases (Guzman et al., 2025). A previous study reported that SMS-1 loss impairs motor neuron-dependent behaviours, such as locomotion and egg-laying, but the underlying mechanisms remained unresolved (Hao et al., 2017). In this study, we re-examined the potential role of SMS-1 in *C. elegans* egg-laying and locomotion by generating independent *sms-1* deletion alleles using CRISPR/Cas9 and performing quantitative behavioural assays and transgenic rescue. Our results reveal that the egg retention and locomotion phenotypes previously observed in the *sms-1(ok2399)* deletion mutant are not caused by SMS-1 loss. Our findings underscore the importance of rigorous genetic validation when analysing gene knockouts.

## RESULTS

### The *sms-1(ok2399)* deletion allele causes egg retention and reduced locomotion

A previous study identified egg-laying and locomotion phenotypes in *sms-1(ok2399)* animals, without dissecting the underlying mechanism (Hao et al., 2017). To investigate this, we attempted to replicate these behavioural phenotypes after backcrossing the *sms-1(ok2399)* allele with wild-type males five times (Fig. 1).

In *C. elegans*, egg-laying is a highly regulated, rhythmic behaviour that is controlled by multiple neuronal and environmental inputs (Fenk and de Bono, 2015; Zhang et al., 2008). Young adult hermaphrodites typically retain ~15 fertilised eggs in their uterus. When regulatory inputs are disrupted, the egg-laying rate may increase or decrease, thus altering the number of eggs retained in the uterus at a defined developmental timepoint (Scharf et al., 2021). We cultivated wild-type and *sms-1(ok2399)* animals in parallel and counted the number of eggs retained in the uterus of 1-day-old adult hermaphrodites (36 h after the mid-L4 larval stage of development). Consistent with previous data, we found that *sms-1(ok2399)* 1-day-old adult hermaphrodites retained more eggs (~35) than wild-type animals (~18) (Fig. 1A,B). To confirm that SMS-1 also controls motor function, we measured locomotion speed of wild-type and *sms-1(ok2399)* 1-day-old hermaphrodites (24 h after the mid-L4 larval stage of development). We found that *sms-1(ok2399)* mutants exhibit reduced locomotion (~51 μm/s) compared with wild-type animals (~111 μm/s), confirming previous findings (Fig. 1C). Together, these

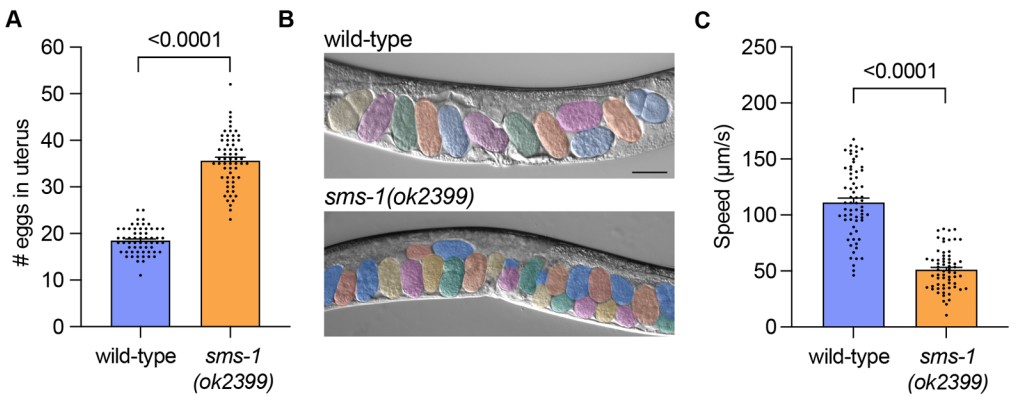

**Fig. 1.** ***sms-1(ok2399)* animals exhibit excess egg retention and reduced locomotion.** (A,B) Quantification (A) and DIC micrographs (B) of wild type and *sms-1(ok2399)* egg retention in 1-day-old hermaphrodites. Eggs are pseudocolored. Worm images are oriented anterior to the left, ventral down. *n*=60. Scale bar: 50 µm. (C) Quantification of wild type and *sms-1(ok2399)* locomotory speed (distance travelled/time) in 1-day-old hermaphrodites. *n*=64-65. Statistical significance was assessed using an unpaired *t*-test (A,C). Error bars indicate s.e.m. Significance was considered at *P*<0.05.

results show that *sms-1(ok2399)* animals exhibit defects in egg-laying and locomotion, consistent with the previous report.

## Overexpressing *sms-1* fails to rescue egg retention and locomotion defects

The *sms-1* gene is predicted to generate a protein containing six transmembrane domains (Hao et al., 2017). The *sms-1(ok2399)* deletion is likely a loss-of-function allele as it is predicted to disrupt splicing and cause a frameshift, resulting in loss of the protein sequence from transmembrane domain 3 onward (Hao et al., 2017).

To assess whether loss of *sms-1* underlies the egg retention and locomotion defects, we attempted phenotypic rescue by reintroducing *sms-1* genomic DNA driven by its endogenous promoter (2 kb upstream of the start codon) into *sms-1(ok2399)* mutant animals (Fig. 2A-C). However, analysis of three independent transgenic lines showed that *sms-1* overexpression failed to rescue either the egg retention or locomotion defects (Fig. 2B,C).

RNA sequencing analysis shows that *sms-1* is predominantly expressed in the hypodermis, with lower-level of expression all other tissues (Cao et al., 2017; Packer et al., 2019). To confirm that the endogenous *sms-1* promoter we used for the rescue experiments drives expression in the correct tissues, we generated a GFP reporter driven by the same promoter (Fig. 2D,E). This reporter recapitulated the expected *sms-1* expression pattern, with strong expression in the hypodermis and other tissues, including the intestine and pharynx (Fig. 2E). The lack of rescue suggested to us that the egg retention and locomotion defects observed in *sms-1(ok2399)* may not be caused by *sms-1* loss and potentially by a background mutation.

## CRISPR/Cas9-generated *sms-1* deletion alleles (*rp398* and *rp399*) exhibit wild-type egg-laying and locomotion

To independently assess the function of SMS-1 in egg-laying and locomotion, we generated two independent *sms-1* alleles using CRISPR/Cas9 (Fig. 3A). In contrast to the *ok2399* allele, which deletes 620 bp spanning exons 3-4, the newly generated alleles, *rp398* and *rp399*, contain larger deletions of 3163 bp and 3086 bp, respectively, that remove the majority of the *sms-1* coding region (Fig. 3A; Dataset 1). These deletions are thus predicted null alleles. We next assessed egg-laying and locomotion behaviours in all three *sms-1* deletion mutants in parallel with wild-type animals. We found that the *sms-1(rp398)* and *sms-1(rp399)* animals exhibited wild-type egg retention and locomotion defects, while *sms-1(ok2399)* were defective (Fig. 3B-D). Together, these results indicate that the

phenotypes previously associated with the *sms-1(ok2399)* allele are not due to loss of *sms-1* function. Thus, we suggest that future studies of SMS-1 function should utilise the *rp398* and *rp399* alleles we generated.

## DISCUSSION

Our results provide a rigorous reassessment of the role of *sms-1* in regulating egg-laying and locomotion in *C. elegans*. Consistent with a previous report, the *sms-1(ok2399)* allele exhibited excess egg retention and impaired locomotion. However, overexpression of the *sms-1* gene driven by its endogenous promoter did not restore normal egg-laying or locomotion, despite confirming promoter activity in the pharynx, intestine, and hypodermis. To further test whether *sms-1* loss underlies these behavioural phenotypes, we generated two independent CRISPR/Cas9 deletion alleles, *rp398* and *rp399*, which remove the majority of the coding sequence and are predicted nulls. These alleles exhibited normal egg-laying and locomotion, in contrast to the *ok2399* allele. Our findings indicate that the phenotypes associated with the *ok2399* allele are not a consequence of *sms-1* loss, indicating that *ok2399* may harbour additional background mutations or allele-specific effects that contribute to the observed defects.

Our study underscores the importance of using multiple independent alleles and genetic validation when linking specific genes to phenotypes. We suggest that future studies investigating SMS-1 function in controlling SM biology, including lipidomics, should employ our CRISPR/Cas9-generated *sms-1* alleles, which we will deposit at the *Caenorhabditis* Genetics Center.

## MATERIALS AND METHODS
### *C. elegans* strains and culture
*C. elegans* hermaphrodites were maintained according to standard protocols at 20°C on nematode growth medium (NGM) plates with *Escherichia coli* (OP50) bacteria as a food source. The wild-type strain used was Bristol, N2. Mutant strains were backcrossed to N2 five times and animals were well-fed for at least two generations before performing experiments. A list of the strains and plasmids used in this study, and source data are provided in Table S1.

### CRISPR-Cas9 genome editing
To generate *sms-1* deletion mutants, adult wild-type hermaphrodites were microinjected with Cas9 protein, tracrRNA, and two crRNAs: 5′ crRNA (CCCTTTAACAGTTGACTCAT), 3′ crRNA (GATGATGTTCCCGGGT-GCTT) from IDT. *sms-1* deletions were identified by PCR and confirmed by

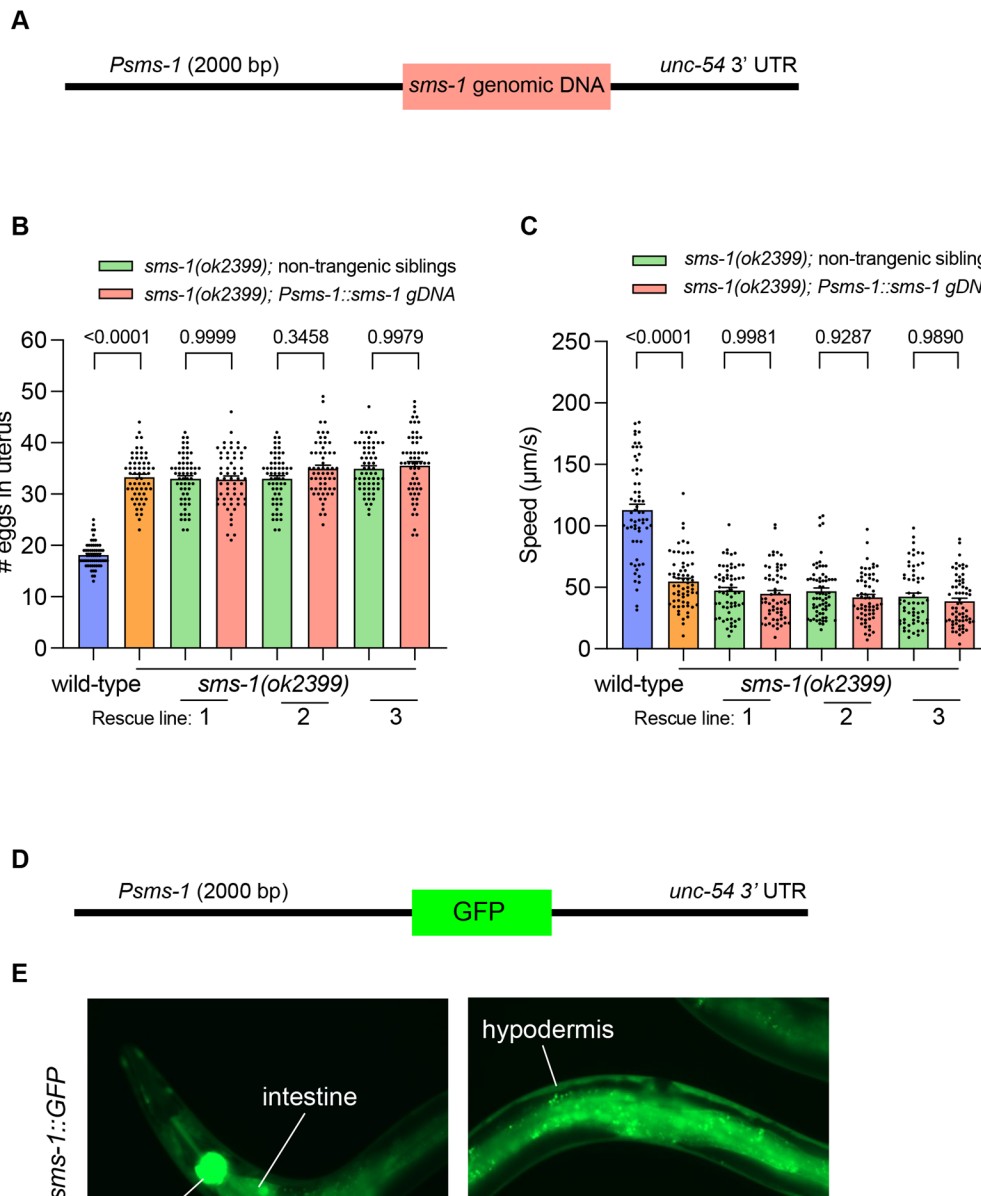

**Fig. 2. Transgenic *sms-1* expression fails to rescue the *sms-1(ok2399)* egg retention and locomotion phenotypes.**
(A) Schematic of the *Psms-1::sms-1-* genomic DNA rescue construct used to rescue the egg retention and locomotion phenotypes of *sms-1(ok2399)* animals. *Psms-1*=2000 bp *sms-1* promoter region; *sms-1* genomic DNA=4679 bp genomic DNA; *unc-54* 3′ UTR=untranslated region.
(B) Quantification of wild-type, *sms-1(ok2399)*, and *sms-1(ok2399); Psms-1::sms-1 gDNA* transgenic animal egg retention in 1-day-old hermaphrodites. Green bars, non-transgenic siblings; pink bars, transgenic animals. *n*=60.
(C) Quantification of wild-type, *sms-1(ok2399)*, and *sms-1(ok2399); Psms-1::sms-1 gDNA* transgenic animal locomotory speed (distance travelled/time) in 1-day-old hermaphrodites. Green bars, non-transgenic siblings; pink bars, transgenic animals. *n*=61-65.
(D) Schematic of the *Psms-1::GFP* fluorescent reporter construct used to examine the where the *sms-1* promoter used in the rescue experiments (A-C) drives expression.
(E) Expression of the *Psms-1::GFP* fluorescent reporter in wild-type animals. GFP expression is detected in the pharynx and intestine (left panel) and hypodermis (right panel). Scale bar: 50 µm. Statistical significance was assessed using one-way ANOVA followed by a Tukey's multiple comparisons test (B,C). Error bars indicate s.e.m. Significance was considered at *P*<0.05.

Sanger sequencing. The *sms-1(rp398)* and *sms-1(rp399)* alleles generated are 3163 bp and 3086 bp deletions, respectively. Both deletion alleles remove most of the *sms-1* gene (Fig. 3A). Each allele was backcrossed to wild-type males prior to analysis. New *sms-1* mutant alleles will be deposited at the *Caenorhabditis* Genetics Center.

### *C. elegans* transgenic strain generation
Reporter gene constructs were generated by PCR amplifying DNA elements and cloning them into Fire vectors. The constructs were injected into young adult hermaphrodites using standard methods.

### *sms-1p::gfp* reporter construct
A 2000 bp sequence upstream of the *sms-1* start codon was amplified from *C. elegans* genomic DNA using forward (5′-TGATTACGCCAAGCTTA-GAGGGCTCTGGAGAAAAACC-3′) and reverse (5′-CCAATCCCGGG-GATCCGTTATGTCACACGGTGTTCG-3′) oligonucleotides incorporating *HindIII-BamHI* restriction sites. The *HindIII-BamHI*-digested promoter fragment was ligated into *HindIII-BamHI*-digested pPD95.75 vector,

resulting in *Psms-1::gfp*, which was injected at 20 ng/µl, with 5 ng/µl *Pmyo-2::mCherry* vector into wild-type animals.

### *Psms-1::sms-1* genomic DNA extrachromosomal array rescue construct
*sms-1(isoform a)* gDNA (4679 bp) was amplified from genomic DNA using forward (5′-AGGACCCTTGGCTAGCATGAAAATGTCTTGGAATCA-TCAA-3′) and reverse (5′-GATATCAATACCATGGTCATTCGAAAGC-AGGTCGTG-3′) oligonucleotides incorporating *NheI-NcoI* restriction sites. The *NheI-NcoI*-digested promoter fragment was inserted using the In-Fusion HD Cloning Kit (Takara Bio) into *NheI-NcoI*-digested *ges-1p:: asah-1 cDNA* vector (to remove *asah-1* cDNA).

A 2000 bp sequence upstream of the *sms-1* start codon was then amplified from *C. elegans* genomic DNA with forward (5′-TGATTACGCCAAG-CTTAGAGGGCTCTGGAGAAAAACC-3′) and reverse (5′-CCAATCCC-GGGGATCCGTTATGTCACACGGTGTTCG-3′) oligonucleotides from incorporating *HindIII-BamHI* restriction sites. The *Pges-1::sms-1* gDNA plasmid was digested using *HindIII-BamHI* to remove the *ges-1* promoter and the *sms-1* promoter was inserted using the In-Fusion HD Cloning Kit

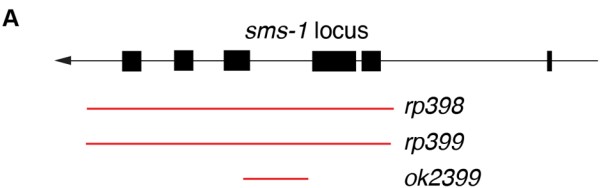

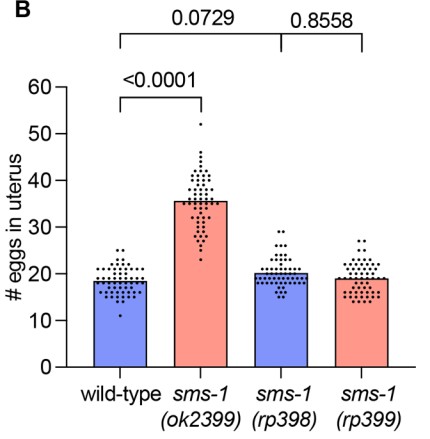

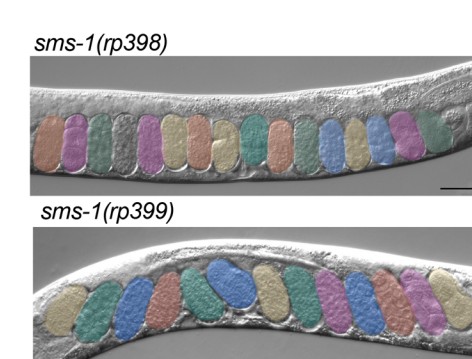

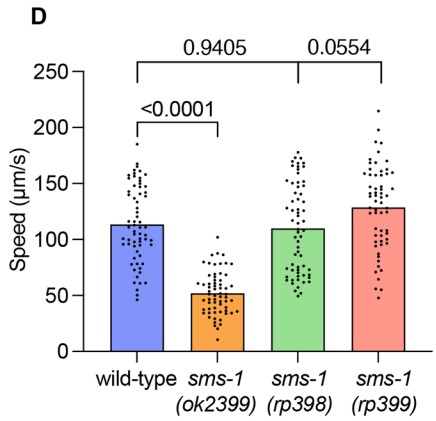

**Fig. 3. CRISPR/Cas9-generated *sms-1* deletion alleles (*rp398* and *rp399*) do not phenocopy the *ok2399* egg retention and locomotion phenotypes.**
(A) Structure of the *sms-1* locus. *ok2399*=620 bp deletion; *rp398*=3163 bp deletion; *rp399*=3086 bp deletion. Black boxes, coding regions; black lines, introns; red lines, deletion alleles. (B) Quantification of wild-type and *sms-1(ok2399, rp398* and *rp399)* egg retention in 1-day-old hermaphrodites. (C) DIC micrographs of *sms-1(rp398* and *rp399)* egg retention in 1-day-old hermaphrodites [compare with wild type and *sms-1(ok2399)* images in Fig. 1B]. Eggs are pseudocolored. Worm images are oriented anterior to the left, ventral down. *n*=60. Scale bar: 50 µm. (D) Quantification of wild type and *sms-1(ok2399, rp398* and *rp399)* locomotory speed (distance travelled/time) in 1-day-old hermaphrodites. *n*=61-64. Statistical significance was assessed using one-way ANOVA followed by a Tukey's multiple comparisons test (B,D). Error bars indicate s.e.m. Significance was considered at *P*<0.05.

(Takara Bio). The resultant *Psms-1::sms-1 genomic DNA* plasmid was injected at 50 ng/µl, with 5 ng/µl *Pmyo-2::mCherry* vector into *sms-1(ok2399)* animals.

### Locomotion analysis
Worm locomotion was analysed at room temperature on 1-day-old adults using WormLab imaging (MBF Bioscience). NGM plates used for tracking were freshly seeded with 20 µl OP50 10 min before use. To standardise acclimation time and food conditions across groups, plates were analysed in a strictly interleaved rotating order until six plates were analysed per group. Six animals were placed on each plate, and their tracks were recorded for 1 min.

### Egg retention analysis
The number of eggs *in utero* were counted in age-synchronised hermaphrodites. Worms were picked at the mid-L4 stage based on the morphology of the vulval invagination. 36 h later, individual worms were lysed in 20 µl of a freshly prepared lysis buffer (1:1 mixture of household bleach and NaOH). Once hermaphrodites dissolved, the released eggs were counted under a dissecting microscope.

### Microscopy/expression analysis
Worms were imaged at the L4 stage to determine the expression pattern of *Psms-1::gfp*. Worms were anaesthetised using levamisole (0.1 ng/µl) and

mounted on 5% agarose pads on glass slides. Fluorescence images were acquired using a Zeiss Axio Imager M2 and Zen software tools. Figures were prepared using ImageJ and Adobe Illustrator.

### Statistics and reproducibility
Statistical analyses were conducted in GraphPad Prism (v9.5). Significance (*P*<0.05) was determined via unpaired Student's *t*-test for two-group comparisons, or ANOVA (with Dunnett's or Tukey's post-hoc tests) for multiple conditions. All experiments were performed with at least three independent biological replicates. No data were excluded, and investigators were blinded to genotype during analysis.

### Acknowledgements
We thank members of the Pocock Laboratory for comments on the manuscript. Some strains were provided by the *Caenorhabditis* Genetics Center (University of Minnesota), which is funded by NIH Office of Research Infrastructure Programs (P40 OD010440).

### Competing interests
The authors declare no competing or financial interests.

### Author contributions
Conceptualization: W.W., R.P.; Data curation: W.W., R.P.; Formal analysis: W.W., X.G.; Investigation: W.W., X.G.; Methodology: W.W., R.P.; Project administration:

W.W., R.P.; Supervision: W.W., R.P.; Validation: W.W.; Visualization: W.W., X.G.; Writing – original draft: W.W.; Writing – review & editing: X.G., R.P.

**Funding**

National Health and Medical Research Council (Senior Research Fellowship GNT1137645 to R.P.) and Veski Innovation Fellowship (VIF23 to R.P.). Open Access funding provided by Council of Australian University Librarians (CAUL). Deposited in PMC for immediate release.

**Data and resource availability**

All relevant data and details of resources can be found within the article and its supplementary information. Source data are in Table S1. There are no accession codes, unique identifiers, or weblinks in our study and no restrictions on data availability. Materials will be available upon request from the Pocock laboratory.

**Peer review history**

The peer review history is available online at https://journals.biologists.com/bio/lookup/doi/10.1242/bio.062520.reviewer-comments.pdf

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
