## [Peer Review File · Biology Open]

Loss of sphingomyelin synthase-1 does not cause egg retention or locomotion defects in *Caenorhabditis elegans*

Wenyue Wang, Xiaoyan Gao and Roger Pocock

DOI: 10.1242/bio.062520

Editor: Sandhya Koushika

Review timeline

Original submission:	10 February 2026
Editorial decision:	18 February 2026
First revision received:	23 February 2026
Accepted:	25 February 2026

Original submission

First decision letter

MS ID#: bio.062520

MS Title: Loss of sphingomyelin synthase-1 does not cause egg retention or locomotion defects in *Caenorhabditis elegans*

Authors: Wenyue Wang, Xiaoyan Gao and Roger Pocock

I have now reached a decision on the above manuscript.

The reviewer reports are shown at the bottom of this email.

As you will see, the reviewers gave favourable reports. I would like you share the sequence data for the crisper alleles and precisely how it alters the protein encoded by these newer alleles you have generated. Have independently sequenced the ok2399 allele and can you share if it affects the coding region of sms-1? These will form a useful addition to the manuscript. Do you think the egg playing phenotype of the ok2399 is closely linked to sms-1? Do ok2399/crisper alleles also have the reported phenotypes? These points would be useful to address in the revision. These might be useful to address in the revision. You do not need to carry out whole genome sequencing of the ok2399 allele.

I hope that you will be able to carry these out, because we would like to be able to accept your paper.

At this stage, we also ask you to ensure your manuscript complies with our formatting guidelines - please see our manuscript preparation guidelines for details. Provided you are able to fully address the referees' comments, we are positive about publication of your paper (we accept over 95% of revision submissions) and therefore hope you won't mind any extra work involved in reformatting your manuscript at this point.

Please upload both a 'clean' version of your Word file, along with a highlighted version clearly showing where you have made changes in the revised manuscript. Please avoid using 'Track changes' in Word files as these are lost in PDF conversion.

I should be grateful if you would also provide a point-by-point response detailing how you have dealt with the points raised by the reviewers in the 'Response to Reviewers' box. Please attend to all of the reviewers' comments. If you do not agree with any of their criticisms or suggestions please explain clearly why this is so.

Reviewer 1

Comments for the author

Overall, the manuscript was pretty clear and concise. It addresses an error in literature and is highly relevant for future studies in the field.

Reviewer 2

Comments for the author

The rationale of the work seems logical. Here are some of the major points needed to be addressed.

1. The authors should perform a whole genome sequencing of sms-1(ok2399) to figure out which background mutation is causing the phenotype? that will be a significant contribution.
2. What is the rationale for selecting the specific mutants ((rp398 and rp399)? is it random?
3. Is there a crosstalk between sphingomyelin with the basement membrane?

Reviewer's Responses to Questions

Experimental quality

Does each figure have the proper controls?

If 'No', please indicate reasons in Comments for Author box below.

Reviewer #1:

- Yes

Reviewer #2:

- Yes

Were the data analyzed using appropriate statistical tests?

If 'No', please indicate reasons in Comments for Author box below.

Reviewer #1:

- Yes

Reviewer #2:

- Yes

Reproducibility

Were experiments performed using adequate number of biological replicates?

If 'No', please indicate reasons in Comments for Author box below.

Reviewer #1:

- Yes

Reviewer #2:

- Yes

Does the methods section provide sufficient detail to permit reproducibility?

If 'No', please indicate reasons in Comments for Author box below.

Reviewer #1:

- Yes

Reviewer #2:

- Yes

Completeness

Are the manuscript's conclusions supported by the data?

If 'No', please indicate reasons in Comments for Author box below.

Reviewer #1:

- Yes

Reviewer #2:

- No

Scholarship

Do the authors cite and discuss the merits of data that would argue for and against their conclusion?

If 'No', please indicate reasons in Comments for Author box below.

Reviewer #1:

- Yes

Reviewer #2:

- Yes

Does the manuscript title & abstract accurately reflect the contents of the manuscript, without hyperbole?

If 'No', please indicate reasons in Comments for Author box below.

Reviewer #1:

- Yes

Reviewer #2:

- Yes

First revision

Author response to reviewers' comments

Reviewer 1: Overall, the manuscript was pretty clear and concise. It addresses an error in literature and is highly relevant for future studies in the field.

No comments required

Reviewer 2: The rationale of the work seems logical. Here are some of the major points needed to be addressed.

1. The authors should perform a whole genome sequencing of sms-1(ok2399) to figure out which background mutation is causing the phenotype? that will be a significant contribution.

Beyond the scope of this work. Our study provides a warning that phenotypes associated with the sms-1(ok2399) allele should be confirmed in the other independent alleles we isolated.

2. What is the rationale for selecting the specific mutants ((rp398 and rp399)? is it random?

These were the only CRISPR deletion mutants obtained. We aimed to delete as much of the sms-1 locus as possible.

3. Is there a crosstalk between sphingomyelin with the basement membrane?

It is unclear why this question is relevant to this study.

Response to editor requests

I would like you share the sequence data for the CRISPR alleles and precisely how it alters the protein encoded by these newer alleles you have generated.

We have now included this in Figure S1.

Have independently sequenced the ok2399 allele and can you share if it affects the coding region of sms-1? These will form a useful addition to the manuscript.

We have sequenced the ok2399 allele and it confirms the deletion shown on wormbase. We have included this sequencing information and the effect on the SMS-1 protein in Figure S1.

Do you think the egg playing phenotype of the ok2399 is closely linked to sms-1?

Potentially. We outcrossed this allele 5x to N2 so the mutation is likely on the same chromosome.

Do ok2399/CRISPR alleles also have the reported phenotypes? These points would be useful to address in the revision. These might be useful to address in the revision.

No exactly sure what you are asking here. We checked the locomotion and egg-laying phenotypes in all three mutants compare to N2 in the manuscript.

You do not need to carry out whole genome sequencing of the ok2399 allele.

Thanks!

Second decision letter

MS ID#: bio.062520R1

MS Title: Loss of sphingomyelin synthase-1 does not cause egg retention or locomotion defects in *Caenorhabditis elegans*

Authors: Wenyue Wang, Xiaoyan Gao and Roger Pocock

I am happy to tell you that your manuscript has been accepted for publication in Biology Open, pending our standard publication integrity checks. It was accepted on 25th February 2026.